# Sentinel Lymph Node Biopsy in Malignant Melanoma of the Head and Neck: A Single Center Experience

**DOI:** 10.3390/jcm12020553

**Published:** 2023-01-10

**Authors:** Marco Rubatto, Franco Picciotto, Giovenale Moirano, Enrico Fruttero, Virginia Caliendo, Silvia Borriello, Nadia Sciamarrelli, Paolo Fava, Rebecca Senetta, Adriana Lesca, Anna Sapino, Désirée Deandreis, Simone Ribero, Pietro Quaglino

**Affiliations:** 1Section of Dermatology, Department of Medical Sciences, University of Turin, Via Cherasco 23, 10121 Turin, Italy; 2Dermatologic Surgery Section, Department of Surgery, Azienda Ospedaliera Universitaria (AOU) Città della Salute e della Scienza, 10126 Turin, Italy; 3Department of Medical Sciences, University of Turin, 10121 Turin, Italy; 4Pathology Unit, Department of Oncology, University of Turin, 10121 Turin, Italy; 5Department of Medical Science, Division of Nuclear Medicine, University of Turin, 10126 Turin, Italy

**Keywords:** melanoma, SLN, sentinel, head and neck, SPECT/CT, head and neck melanoma

## Abstract

Purpose: This study evaluated the characteristics of patients with head and neck (H&N) melanoma who underwent sentinel lymph node biopsy (SNLB) and assessed the clinical course of patients categorizing subjects according to SLNB status and melanoma location (scalp area vs. non-scalp areas). Methods: Patients undergoing SLNB for melanoma of H&N from 2015 to 2021 were prospectively characterized according to sentinel lymph node (SLN) status. SPECT/CT had been previously performed. Patients were followed until the first adverse event to evaluate progression-free survival. Results: 93 patients were enrolled. SLNB was negative in 75 patients. The median Breslow index was higher for patients with positive SLNB compared with patients with negative SLNB. In addition, the Breslow index was higher for melanoma of the scalp compared with non-scalp melanoma. The median follow-up was 24.8 months. Progression occurred at the systemic level in the 62.5% of cases. There was a significant association between positive SLNB and progression (*p*-value < 0.01) of disease, with lower progression-free survival for patients with melanoma of the scalp compared with those with melanoma at other anatomic sites (*p*-value: 0.15). Conclusions: Scalp melanomas are more aggressive than other types of H&N melanomas. Sentinel lymph node status is the strongest prognostic criterion for recurrence.

## 1. Introduction

The prevalence of head-and-neck (H&N) melanoma has been increasing in recent decades, with melanomas of head and neck accounting for 25% of all cutaneous melanomas [1]. The most common subtypes of H&N melanoma include superficial spreading, nodular, lentigo maligna, and desmoplastic melanoma [2]. Lentigo maligna is more commonly found in the H&N compared to other sites [3].

Features such as Breslow tumor thickness, advanced Clark level, presence of tumor ulceration, younger age, anatomic site, and histological subtype are associated with lymph node metastasis [4]. 

It appears that primary melanomas arising in the H&N district have a worse prognosis than those located in other sites [5]. Estimated 5-year overall survival is 74% for melanoma of the H&N, 84% for melanoma of the extremities, and 82% for melanoma of the trunk [6]. Mortality rates among H&N melanomas differ by site: lesions of the scalp and neck have the highest mortality, with a 10-year survival of 60% [7]. Occult lymph node metastasis is present in 15% to 20% of patients with melanoma of the H&N and clinically-negative lymph nodes [8]. 

Sentinel lymph node biopsy (SLNB) can identify patients with intermediate and thick melanoma who are at risk of occult node metastases [9,10,11]. Concerning thin melanoma, there is still debate if they should be submitted to SLNB or not independently of other prognostic features. The most recent guidelines suggest performing SLNB from stage IB [12]. For patients with stage IA with melanomas between 0.76 and 1.0 mm thick without ulceration, SLNB should be considered in appropriate clinical setting [13]. SLNB may be challenging in the H&N region mainly due to the regional course of cranial nerves and lymphatic drainage. Aberrant drainage patterns are observed in 24–70% of H&N melanomas, and multiple drainage basins are observed in 50% of cases [14]. Several studies have been conducted to identify the possible benefit of SLNB in this anatomic site. The results of MSLT-I confirmed that SLNB with immediate lymphadenectomy reduces the risk of lymph node recurrence, distant metastasis, and death from melanoma. However, the German dermatologic cooperative oncology group selective lymphadenectomy (DeCoG-SLT) study and MSLT-II concluded that immediate complete lymph node dissection (CLND) improved the regional control rate, but did not impact disease-specific survival among patients with stage IB-IIC melanoma with positive SLNB [14]. 

Since there is evidence that the location of H&N melanoma is an independent adverse prognostic factor, the conclusions of these two trials should not be generalized to this specific group [14]. The conclusion of these trials cannot be applied to H&N melanoma due to the prognostic implications of this anatomical site. In the MSLT-II trial, 13.7% of patients had H&N melanoma: subgroup analysis showed no significant difference between CLND versus observation following a positive SLNB in these patients [15]. Following the results of the MSLT II, sentinel lymph node (SLN) status is not an indication to CLND, but it does give an indication to adjuvant therapy, which with new effective drugs increases its importance.

With advances in technology, SPECT-CT provides detailed anatomic localization of SLNB and allows detection of more SLNBs than conventional planar lymphoscintigraphy (PL). Routine use of SPECT-CT is recommended to optimize detection and localization of SLNBs in patients with melanoma of the H&N [16]. 

The primary objective of the study was to evaluate the clinical characteristics of patients with H&N melanoma who underwent SNLB. In addition, we evaluated the clinical course of patients categorizing subjects according to SLNB status (positive vs. negative) and melanoma location (scalp area vs. non-scalp areas).

As a secondary objective, lymph node drainage sites were evaluated based on the location of the primary melanoma.

## 2. Methods

Data on primary cutaneous melanoma in the H&N region, without distant metastasis at time of diagnosis, and undergoing SLNB were prospectively collected from 2015 to 2021 at the Department of Dermatology and dermatologic surgery of Turin University hospital. Data on gender, age, primary tumor site, Breslow index, level of Clark, and ulceration were collected at baseline. 

TNM staging, clinical stage, and sentinel lymph node screening were assessed through SLNB. According to the AIOM guidelines [12], SLN screening is indicated in patients without detectable lymph node or distant metastasis in which the Breslow index was >1.0 mm or <1.0 mm in the presence of ulceration. Patients underwent pre-operative mapping of draining lymph nodes the day before surgery. Pre-operative PL for sentinel lymph node mapping was performed after circumferential intradermal injections of ^99m^Tc labeled manocolloid (20–45 MBq) equally subdivided in four syringes, in a volume of approximately 0.10 mL, around the lesion or the excisional biopsy site. Dynamic flow imaging of H&N over 30 min was performed to identify regional lymphatic basins and to differentiate true SLNs from non-SLNs. Conventional PL was completed by SPECT/CT of the same region to provide accurate anatomical localization of SLNs and, in some cases, to demonstrate SLNs not detected on planar images. SPECT/CT is a new hybrid technique that fuses functional scintigraphic images with anatomical information collected by CT. In order to determine the lymph node drainage, we divided the facial neck district in 13 anatomical sites (anterior scalp, coronal scalp, posterior scalp, upper face, lower face, nose, ear, preauricolar, anterior upper neck, coronal neck, posterior upper neck, anterior lower neck, posterior lower neck). For statistical analysis, we divided patients by considering scalp melanomas as one group and other H&N anatomical districts as another group. Neck lymph nodes were tagged according to 13 groups. These groups included 6 laterocervical lymph nodes groups (categorized according to Robbins’ levels; level I: submental and submandibular group; level II: upper jugular group; level III: middle jugular group; Level IV: lower jugular group; level V: posterior triangle group; level VI: anterior compartment) as well as preauricular, periparotid, intra-parotid retroauricular, mastoid, occipital, supraclavicular groups In patients with histologically-confirmed lymph node metastasis, lymph node dissection was performed according to current year guideline. 

Patient were followed until the first adverse event to evaluate progression-free survival (defined as time from date of SLNB until disease progression or death). 

All statistical analyses were carried out using Stata software, version 17.0. The values obtained when studying the quantitative variables were described using median and interquartile range (IQR), while for qualitative variables absolute and relative frequencies were used. Demographic and clinical variables were compared according to the SLNB status (positive vs. negative) and anatomic site of the primary lesion (scalp vs. non-scalp). Quantitative variables were compared using the Mann-Whitney test. The Chi-square test compared qualitative variables, using Fisher’s test when necessary. Kaplan–Meier curves were constructed to disclose differences in progression-free survival analyses and log-rank was applied when comparing them. Lastly, Cox regression models were fitted to understand potential influence of baseline covariates on the progression-free survival. A level of statistical significance of less than 5% (*p* < 0.5) was adopted.

## 3. Results

A total of 93 patients were enrolled with primary melanoma of the H&N area who underwent SNLB between 2015 and 2021 after previous SPECT-CT evaluation. 

There were 40 (43%) adult females and 53 (57%) adult males. The most common tumor localization was the face (38.0%), followed by the scalp (31.5%), ears (20.7%), and neck (9.8%). Median Breslow thickness value was 1.8 mm (IQR range: 1.1–3.0 mm). SLNB was negative in 75 patients (35 patients were Ib stage (46.7%), 21 in IIa stage (28%), 12 in IIb stage (16%), 7 in stage IIc (9.3%); 18 patients showed SLN involvement 5 in IIIa (28%), 5 in IIIb (28%), 7 in IIIc (38%), and 1 in IIId (6%). The majority of patients (86%, *n* = 79) drained unilaterally. Bilateral drainage occurred in 13 patients (14%) and the majority of these were from the scalp (*n* = 6, 46%). Among all lymph node groups, upper jugular group (Robbins level II) was the most common site of SLNs in our cohort (63%, *n* = 59). Melanoma from all anatomical sites drained to Robbins level II, with the greatest percentage of drainages arising from ear melanoma. The other most common sites of drainage were the submental and submandibular group (Robbins level I) (30%, *n* = 30), receiving drainage most commonly from the face.

Table 1 reports the demographic and clinical characteristics according to SLNB status. The median Breslow index was higher for patients with a positive SLNB compared with patients with a negative SLNB (2.2 mm; IQR range: 1.8–5.0 mm vs, and 1.8 mm; IQR range: 1.1–3.0 mm). No difference in terms of ulceration was observed between SLNB-positive and SLNB-negative patients. Among the 18 patients who had a positive SLN, 9 did not have a complete dissection because AIOM guidelines from 2020 no longer recommend it. More specifically, among the 18 SLNB positive subjects, 1 patient had melanoma in anterior scalp and the sentinel lymph node was located in the retro-auricular site, 4 patients had melanoma at the coronal scalp and the sentinel was positive at II and V Robbins levels, and 2 patients with posterior scalp melanoma had both positive sentinel node at V Robbins level. Five patients with melanoma in upper face had a positive sentinel lymph node at levels I and II, while three patients with melanoma in lower face had SLNB+ at level I. In one patient with melanoma of the ear, a metastatic lymph node was identified at the second latero-cervical level. One preauricular melanoma drained to the periparotid level and 1 coronal neck melanoma drained to Robbins level II. In addition, Table 1 also shows the demographic and clinical characteristics of patients with melanoma in the scalp (*n* = 29) vs. patients with melanoma in another region of the H&N (*n* = 64). The Breslow index was higher for melanoma of the scalp melanoma compared with non-scalp melanoma (median index: 2.5 mm; IQR range: 1.4–4.2 mm vs, and 1.8 mm; IQR range: 1.2–2.6 mm). No difference was observed for the proportion of ulcerated lesions or for the proportion of positive SLNB. 

Of the 93 patients who underwent SLNB, we follow 82 subjects for progression-free survival. The median follow-up for the cohort was 24.8 months (IQR: 12.6–35.4) with 19 events observed. Progression occurred at the systemic level in 62.5% of cases, at lymph node level in the 25% of cases and at pericicatricial level in the 12.5% of cases. Kaplan-Meier analysis showed a significant association (*p* = 0.01) between positive SLNB and progression (log-rank test *p*-value < 0.01) (Figure 1A). Kaplan–Meier analysis (Figure 1B) suggested a lower progression-free survival for subjects with melanoma of the scalp compared to subjects with melanoma in other anatomic sites, although the difference was not statistically significant (log-rank test *p*-value: 0.15). In Table 2, hazard ratios (HR) and 95% confidence intervals (CIs) obtained from univariable and multivariable Cox regression Models are reported. 

Of all 93 patients enrolled, 7 received adjuvant therapy (7.55%), of which 4 had immunotherapy and 3 had targeted therapy depending on BRAF mutational status. Four patients died during the observation period: one had positive lymph nodes at SNLB and three had negative lymph nodes. 

## 4. Discussion

H&N melanoma has always been a debated topic. H&N melanomas constitute a distinct variety of cutaneous melanomas, biologically behaving more aggressively and being diagnosed with a temporal delay. At presentation, approximately 85% of patients with H&N have localized disease, 10% have regional disease, and 5% have systemic disease [17].

It is known from literature that H&N melanomas have higher relapse rates than those of other body districts [18]. The relapse rate is estimated to be around 13% for scalp lesions compared to 5% for lesions of remaining body areas [18].

A crucial step in the staging of H&N melanoma is the search for sentinel lymph nodes. The risk of finding a positive sentinel lymph node is directly proportional to thickness of the melanoma, ulceration, and to the number of mitoses [19] Sentinel lymph node status is a powerful predictor of prognosis in cutaneous melanoma, with higher rates of recurrence and worse disease-specific and overall survival in SN-positive patients [20].

CLND is still a debated topic, especially in such a delicate district as the H&N. 

Therapeutic benefit of CLND is based on the theory that occult metastases in the nodal basin after SLNB may lead to local or systemic progression [14]. However, several studies have shown that most recurrences in SLNB-positive patients are systemic regardless of CLND, which negates the value of additional lymph node dissection [21]. In fact, CLND has not been shown to be associated with better OS in cutaneous H&N melanoma patients with SNLB+ [14]. However, the Multicenter Selective Lymphadenectomy Trial II (MSLT-II) found that immediate CLND has better regional control than observation alone (92% vs. 77%, *p* < 0.001). Regional control in the H&N may be more important than other anatomic sites given the critical structures within this region, including the trachea, esophagus, cranial nerves, and major vessels. Adverse events most commonly involved with lymph node dissection include lymphedema, wound dehiscence, and infection [22].

The role of the SLN has become even more important since 2019, since adjuvant therapy was approved for patients from stage IIIa. Target therapy and immunotherapy have demonstrated to decreases residual micrometastatic disease that may be source of future relapse [23].

As mentioned, H&N melanomas constitute a distinct variety of cutaneous melanomas, behaving biologically more aggressively and being diagnosed with a temporal delay [7]. Precisely because of diagnostic delay, they have a higher Breslow at diagnosis, which is a negative prognostic factor. In addition, young people seem to be more likely to have nodal recurrence, probably because of the site being hidden by hair. 

The parotid site constitutes a delicate surgical site because of the risk of facial nerve damage, salivary fistulae, or Frey’s syndrome. In our cases, we observed adverse events following surgery in this anatomic district. Olilla et al. [24] reported lymphatic drainage at the parotid level similar to our case series, and in particular drainage most commonly occurred from forehead, cheek, and pre- auricular melanoma.

Level II of Robbins was the most common site of SLNS in our cohort. This is consistent with reports by Lin et al. [25] and Jensen et al. [26].

Regarding scalp melanomas, they typically exhibit more aggressive biological behavior and are often diagnosed at a late stage [18]. An association between scalp melanoma and older age is reported in the literature, particularly for an increased occurrence of baldness in the elderly. In fact, it has been demonstrated that there is an up to 7-fold increased risk of scalp melanoma in the population with moderate to severe baldness compared to non-bald individuals [27]. Moreover, melanomas of the scalp are typically associated with an increase in Breslow thickness at initial presentation [18], probably due to a diagnostic delay caused by a site that is difficult to explore due to the presence of hair and therefore not easily accessible to autonomic or clinical examination [28]. In fact, the first clinical symptoms generally appear when the melanoma has already reached significant thickness.

Since 2015, SPECT/CT has been the first choice for correct sentinel lymph node localization and correct staging of patients [29]. This tomographic technique can be useful in achieving better lymph node detection and their correct anatomical localization in H&N melanoma, increasing the diagnostic accuracy of SLNB. According to a recent literature review, the number of SLNs detected by SPECT/CT and PL (conventional lymphoscintigraphy) showed a higher, statistically significant number of SLNs in melanoma of the neck between the two techniques with an OR of 1.13 in favor of SPECT/CT (95% CI: 1.06–1.2; *p* < 0.001, random-effects model) and moderate heterogeneity (I2 = 60.08%) [30] SPECT/CT has two advantages over conventional lymphoscintigraphy: it provides anatomical localization of SLNs and is useful to demonstrate SLNs that are not detected on planar images. Visualization of lymph nodes in relation to anatomic structures facilitates interpretation and optimally prepares the surgeon. SPECT/CT can be considered as a surgical “road map” in a district with such complex anatomy, a high number of lymph nodes, and sometimes, unpredictable drainage patterns. In fact, it allows for a more precise surgical procedure, reducing operating time and minimizing morbidity from complications. In the literature, as well as our experience, thanks to SPECT-CT an additional SLN sentinel lymph node can be detected in 16% of patients [31].

## 5. Conclusions

The H&N deserves special attention, with close follow-up and careful surgical procedure with adequate margins. In addition, scalp melanomas should be followed with even more attention since they are more aggressive. SLN status is the strongest prognostic criterion for recurrence and survival and thanks to advent of SPECT-CT greater accuracy in sentinel node detection has been achieved. With the advent of new therapies and the possibility of providing adjuvant therapy to stage III patients, SLNB appears to be a crucial moment in clinical staging. Thus, making the technique more precise is mandatory. In addition, further studies on SLNB and even more on indications to CNLD will need to be performed.

## Figures and Tables

**Figure 1 jcm-12-00553-f001:**
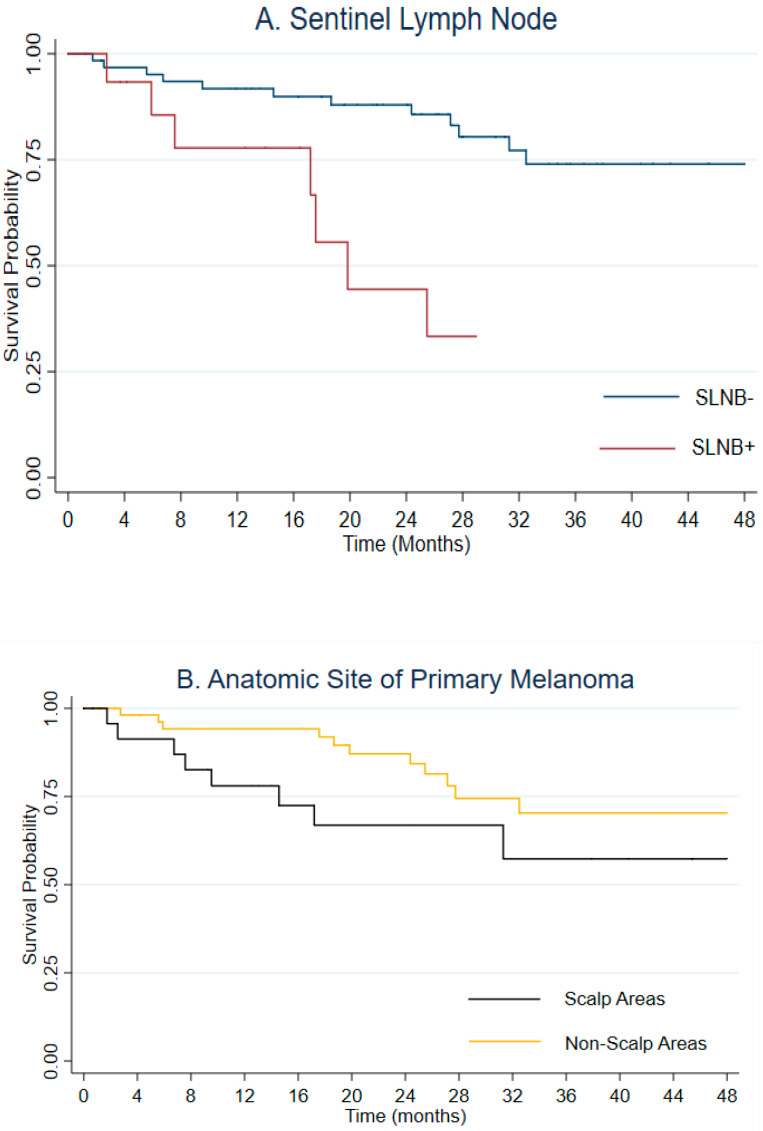
Progression free survival of H&N Melanoma stratified by sentinel lymph node status (panel (**A**)) and anatomic site (panel (**B**)).

**Table 1 jcm-12-00553-t001:** Demographic and clinicopathological characteristics of the study population, stratified by sentinel lymph node biopsy status and anatomic site.

		Nodal Status			Anatomic Site		
Variable	Overall (N = 93)	Nodal Negative (N = 75)	Nodal Positive (N = 18)	*p* ^a^	Non-Scalp (N = 64)	Scalp (N = 29)	*p* ^a^
**Age, median (IQR)**	58 (50–70)	59 (51–71)	52 (41–61)	0.04	53 (45–69)	63 (55–72)	0.01
Sex							
Male	45 (51.6)	40 (53.3)	8 (44.4)	0.49	31 (48.4)	17 (58.6)	0.36
Female	48 (48.4)	35 (46.7)	10 (55.6)		33 (51.6)	12 (41.4)	
**Breslow, median (IQR)**	1.8 (1.2–3.5)	1.8 (1.1–3.0)	2.25 (1.8–5.0)	0.04	1.8 (1.2–2.6)	2.5 (1.4–4.2)	0.04
**T Stage**							
T1	13 (14.3)	13 (17.3)	0 (0.0)	0.10	10 (15.6)	3 (11.1)	0.33
T2	43 (47.3)	35 (46.7)	8 (44.4)		33 (51.6)	10 (37.0)	
T3	15 (16.5)	12 (16.0)	3 (16.7)		10 (15.6)	5 (18.5)	
T4	20 (22.0)	13 (17.3)	7 (38.9)		11 (17.2)	9 (33.3)	
Missing	2	2	0		0	2	
**Ulceration**							
Absent	57 (66.3)	45 (66.2)	12 (66.7)	0.97	40 (67.8)	17 (63.0)	0.66
Present	29 (33.7)	23 (33.8)	6 (33.3)		19 (32.2)	10 (37.0)	
Missing	7	7	0		5	2	
**Anatomic site**							
Scalp	29 (31.5)	22 (29.7)	7 (38.9)	0.63	/	/	/
Face	35 (38.0)	27 (36.4)	8 (44.4)		/	/	
Ear	19 (20.7)	17 (23.0)	2 (11.1)		/	/	
Neck	9 (9.8)	8 (10.8)	1 (5.6)		/	/	
**Drainage**							
Monolateral	79 (86.0%)	64 (86.5)	15 (83.3)	0.49	56 (88.9)	23 (79.3)	0.33
Bilateral	13 (14.1%)	10 (13.5)	3 (16.7)		7 (11.1)	6 (20.7)	
Missing	1	1	0		1	0	
**Nodal Disease**							
Yes	75 (80.6)	/	/	/	53 (82.0)	22 (75.9)	0.43
No	18 (19.4)	/	/		11 (18.0)	7 (24.1)	

^a^*p* value obtained from Wilcoxon-Mann-Whitney or Chi-squared, as appropriate. (IQR: interquartile range).

**Table 2 jcm-12-00553-t002:** Univariate Cox Regression Analysis on relapse-free survival.

Variable	HR ^1^	95% CIS ^1^	HR ^2^	95% CIS ^2^
**Age (1 year increase)**	1.04	1.00–1.08	1.04	1.00–1.08
**Sex**				
Females	Ref	Ref	Ref	Ref
Males	1.85	0.72–4.71	/	/
**Breslow (1 mm increase)**	1.40	1.20–1.63	1.29	1.03–1.65
**T stage**				
T1,T2	Ref	Ref	Ref	Ref
T3,T4	4.09	1.54–10.8	1.20	0.31–4.60
**Ulceration**				
No	Ref	Ref	Ref	Ref
Yes	1.23	0.45–3.39	/	/
**SLNB**				
Negative	Ref	Ref	Ref	Ref
Positive	4.90	1.80–13.30	2.65	0.88–7.95
**Scalp**				
Non scalp	Ref	Ref	Ref	Ref
Scalp	1.92	0.77–4.78	1.31	0.47–3.63

^1^ Crude Cox Regression Model, ^2^ Adjusted Cox Regression Model. (SLNB: Sentinel Lymph Node Biopsy).

## Data Availability

The data that support the findings of this study are available for the corresponding author, MR, upon reasonable request.

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
