# Peer review of "Sentinel Lymph Node Biopsy in Malignant Melanoma of the Head and Neck: A Single Center Experience"

_jcm, 2023, doi:10.3390/jcm12020553_

Round 1

Reviewer 1 Report

Although the paper by Rubatto et al. sought to investigate a highly relevant and current theme, with high translational potential, some issues must be solved before it can be accepted and published:

On pages 3 and 4 there are underlined words?

I do not think that figures and tables conform to the indications for authors.

Page 2, row 83: references are not written correctly

There are points put in front of the bibliographic reference and the introduction in text of tables/figures is also wrong.

All bibliographic references do not comply with the publication rules.

Also, a short text about the viruses and tumors with these localization  would be useful to facilitate the readers comprehension for the subject.

Author Response

Updated

Reviewer 2 Report

Dear authors,

You have presented a relative small prospective unicentric cohort of H&N melanoma pts that underwent SLNB. Results seem all to be in accordance with the literature, which is no demerit and therefore shouldnt preclude its publication. My main concerns with your paper are:

.Abstract is too big! be more concise!

.First line of methods: without what?

.2nd line of methods: what was collected? data?

.You mention a ROC curve regarding mitotic index, but this is not in the results.

.There are several editorial issues with figures 1 and 2; not everybody will know what scalp and nodal 0 and scalp and nodal 1 mean. Review the number indicated in the figures.

.In table 3 it is not clear what are the reference values; e.g. for sex, is it male or female? For breslow? Always indicate the reference and the comparator.

."CLND" meaning is not exploded anywhere in the MS.

.Line 266-267 of the discussion makes a very hard statement; soft the tone here as higher breslow at diagnosis could also be due to higher proliferation index or more aggressive biology.

.What is the reference for line 288??

Author Response

updated

Reviewer 3 Report

This manuscript aims to analyze the role of sentinel node biopsy in melanoma of head and neck. The topic is not novel as several lines of evidence have already shown the role of sentinel node biopsy in melanomas including those of head and neck. The quality of writing is low as the abstract is too long. In addition not clear aim is evident in both abstract and manuscript. The manuscript just described the clinico-pathological characteristics of melanomas of head and neck who underwent pathological esaminations of sentinel node biospy. Even more there are major issues and methodological concerns. The major bias and limitations are represented by the analysis of recurrence free survial and disease free survial analysis since: 1) it is not clear the median follow up of patient population (study ends in 2021 and actually we are in 2022: how patients from 2021 have a 24 months follow up? 2) surgical procedures of lymphoadencetomy changes during the osservational time 3) adjuvant treatment also changes and not all lymphonode positive patients received an adjuvant treatment 4) major prognostic biomarker do not correlate with each other : i.e. tickness and ulceration do not correlate with positivity to the sentinel node status (as a result patient population appears not representative) 4) it is not clear which stage system edition is utilized and why patients with a positive node status can be in stage  Ib 5) authors should be better specified the patients population 6) methodology is not appropriate. Lastly the manuscript requires an extensive editing as 1) there are several mistakes in punctuation 2) some sentences are incomplete 3) references are not in an appropriate format, some of them are not in a citation format, some are missing; 2) several incosistencies are described i.e. there is a figure which is not clear what it refers (see kaplan meier curve by scalp), 3)SPET-CT is reported as SPET-TC 4)  quality of figures is very low 5) table are also in a low quality (i.e. variabile should be read variable).  

Author Response

updated

Round 2

Reviewer 2 Report

fine to publish after amendments

Reviewer 3 Report

The authors have deeply revised the manuscript based on preovious comments. It can be accepted for publication.